# High Efficiency Fluorinated Oligo(ethylenesuccinamide) Coating for Stone

**Mara Camaiti [1,*], Villiam Bortolotti [2], Yijian Cao [1,3], Alessandra Papacchini [4], Antonella Salvini [4] and Leonardo Brizi [5,6,*]** 

1   CNR-Institute of Geosciences and Earth Resources, 50121 Florence, Italy; yijian.cao@nwpu.edu.cn
2   Department DICAM, University of Bologna, 40131 Bologna, Italy; villiam.bortolotti@unibo.it
3   Institute of Culture and Heritage, Northwestern Polytechnical University, Xi'an 710072, China
4   Department of Chemistry, University of Florence, 50019 Sesto Fiorentina, Italy; a_papacchini@alice.it (A.P.); antonella.salvini@unifi.it (A.S.)
5   Department of Physics and Astronomy, University of Bologna, 40127 Bologna, Italy
6   INFN, Sezione di Bologna, 40127 Bologna, Italy
*   Correspondence: mara.camaiti@igg.cnr.it (M.C.); leonardo.brizi2@unibo.it (L.B.);
    Tel.: +39-055-2757558 (M.C.); +39-051-2095163 (L.B.)

**Abstract:** The protection of stone cultural assets is related to the transformation of the surface characteristic from hydrophilic to hydrophobic/superhydrophobic through the application of a coating. The suitability of a coating depends not only on its capability to dramatically change the surface wettability, but also on other parameters such as the modification of kinetics of water absorption, the permanence of vapor diffusivity, the resistance of the coating to aging and the low volatile organic compound emissions during its application. In this work, an oligo(ethylensuccinamide) containing low molecular pendant perfluoropolyether segments (SC2-PFPE) and soluble in environmentally friendly solvents was tested as a protective agent for historic stone artifacts. Magnetic resonance imaging and relaxometry were employed to evaluate the effects of the surface wettability change, to follow the water diffusion inside the rock and to study the porous structure evolution after the application of SC2-PFPE. A sun-like irradiation test was used to investigate the photo-stability of the product. The results demonstrate that the highly photo-stable SC2-PFPE minimizes the surface wettability of the stone by modifying the water sorptivity without significantly affecting its porous structure and vapor diffusivity. The improved performance of SC2-PFPE in comparison to other traditional coatings makes it a potential candidate as an advanced coating for stone cultural heritage protection.

**Keywords:** perfluoropolyethers; cultural heritage; stone protection; oligo(ethylensuccinamide); magnetic resonance imaging; magnetic resonance relaxometry; sorptivity; photo-stability



## 1. Introduction

The preservation of historical buildings and outdoor cultural assets should be a socio-economic priority because they are the historic-cultural testimony of our past, as well as economic resources for the present time [1]. Stone is one of the most important porous materials for most of these artifacts. The main causes of stone degradation are linked to chemical-physical processes involved in the ingress and diffusion of water (liquid or vapor) into the porous structure [2,3]. Indeed, water in the condensed phase dissolves $CO_2$ and pollutants from the atmosphere causing acidic corrosion of the stone and/or its binder [4–8], and it is responsible for internal mechanical stresses caused by hydric dilation [9], freezing-thawing cycles [10,11] and/or salt crystallization [8,12–14].

The use of hydrophobic compounds, both natural compounds [15,16] and synthetic polymers (e.g., acrylics, silicon-based resins, epoxy resins), is a common practice to protect stone surfaces [3,17–19]. However, it has been demonstrated that the efficiency and

durability of the treatments depend on the characteristics of the compounds used, on the treatment procedure and on the chemical-mineralogical nature of the substrate [20–22]. In particular, for effective preservation of the substrate, the protective agent must have high chemical, thermal and photo-oxidative stability, high adhesion to the mineral substrate (interaction but no reaction with stone), low surface tension and molecular sizes which allow a uniform distribution, good penetration into the porous structure, and low propensity for pore blockage [20,23,24]. Last but not least, the protective compound must be soluble in benign solvent for operators and the environment. Among the protective agents used to protect historic stone surfaces, fluorine-containing compounds (e.g., perfluoropolyethers and their derivatives, fluoroelastomers) can be considered potential candidates as excellent protective agents for their higher stability and lower surface tension compared with not fluorinated compounds [25]. Due to the high stability of perfluorinated compounds (chemically very stable and resistant to biodegradation), the drawback of perfluoropolyethers is that they can bioaccumulate with a possible risk of persisting in the environment [26].

Perfluoropolyethers, and the best performing amidic derivatives, highly viscous and low molecular weight oils, can also penetrate deeply into the rock without blocking the pores [27,28]. However, their solubility in only chlorofluorocarbons (CFC) and supercritical $CO_2$ [29], made them not usable as protective agents for historical stone artifacts since 1995. On the other hand, partially fluorinated polymers, such as fluoroelastomers [30] or partially fluorinated acrylic-methacrylic polymer [31–34], show good solubility in common organic solvents, but the size of the macromolecules (medium-high average molecular weight) does not facilitate the penetration and homogeneous distribution of the polymer in the porous structure. As in the case of other polymeric materials, the big molecules can be responsible for the deposition of a superficial film with partial or total pore blockage [20]. The advantage of a good penetration can also be obtained with siloxanes (monomers or small oligomers), and excellent hydrophobic properties can be achieved if mixed with nanomaterials (e.g., nanosilica) [35]. However, due to the in-situ polymerization of the coating, possible reactivity with the rock and insolubility of the final coating (i.e., loss of reversibility) is expected.

In recent publications, we reported advanced results obtained with some low average molecular weight oligoamides containing short pendant perfluoropolyether segments [36–39]. In particular, we found that the oligo(ethylensuccinamide) grafted with short perfluoropolyether (SC2-PFPE) provides superhydrophobic and near superoleophobic properties to highly porous stone surfaces due to the low surface tension of the fluorinated segments [38]. Moreover, the low molecular weight facilitates its diffusion inside the rock, while the polar groups (several amidic groups per molecule) make possible the dissolution in hydro-alcoholic solvents. The amidic groups are also responsible for a strong interaction between SC2-PFPE and the stone, which provides the protective treatment with better resistance to ageing [36].

To emphasize the advanced properties of SC2-PFPE, in this work the photo-stability of the protective agent, as well as the decrease of water sorptivity and the negligible modification of the porous structure of the coated stone specimens were proved. A highly porous biocalcarenite (Lecce stone) was exploited to test the performance of both SC2-PFPE and a highly fluorinated commercial product, taken as reference coating. A sun-like irradiation test was used to investigate the photo-stability, while magnetic resonance imaging (MRI) was employed to study water imbibition and consequently to determine the sorptivity (MRIsorp). Although several techniques can be employed for monitoring the dynamic process of water transport in the porous medium (e.g., neutron radiography, gamma ray, X-ray computed tomography) [40–43], MRI was also proved to be a valid technique for monitoring the water absorption in materials and objects of interest to cultural heritage, with and without conservation treatments [20,21,44–51]. The analysis of the MRI images, taken at increasing intervals of time during capillary water absorption, by an in-house software which implements an algorithm able to accurately identify the height reached by the advancing water-front, even with very low signal, was previously proved to

provide an objective criterion for determining the sorptivity [50]. As the sorptivity of a stone sample is related to the properties and effectiveness of the applied protective coating, this procedure provides a valid parameter to perform objective discrimination among different treatments. Finally, nuclear magnetic resonance relaxometry (NMRR) measurements were performed by an NMR single-sided device (NMR-MOUSE) to characterize the pore space of coated and uncoated stones. In particular, exploiting the magnetic field gradients of NMR-MOUSE, the molecular self-diffusion of water molecules in confined pores can be investigated through the NMR longitudinal and transverse relaxation times $T_1$ and $T_2$, as they are related to the pore structure [49,51–54].

## 2. Materials and Methods

### 2.1. Materials and Coating Procedure

SC2-PFPE, with average $M_w$ = 2050 g/mol, was synthesized in two subsequent steps via condensation reactions as described in [36] and shown in Scheme 1. To dissolve SC2-PFPE in environmentally friendly solvents, e.g., 2-propanol/$H_2O$ 70/30 (*w/w*) [36], a temperature of about 70–80 °C is necessary, however it is possible to apply the coating solution or dispersion even at room temperature. In particular, SC2-PFPEsol, which contains SC2-PFPE 0.5% (*w/w*) in 2-propanol/$H_2O$ 70/30 (*w/w*), was applied as a solution while SC2-PFPEsusp, which contains SC2-PFPE 1% (*w/w*) in 2-propanol/$H_2O$ 70/30 (*w/w*), was applied as a dispersion with soluble and insoluble product. The highly fluorinated commercial product (N215, Poly(hexafluoropropene-*co*-vinylidene fluoride)), employed as an aggregating and protective coating for stone historic surfaces, is a fluoroelastomer with an average Mw = 125,000 g/mol. It was kindly supplied by Solvay-Solexis (Milan, Italy) and was used as a reference. A biocalcarenite (Lecce stone), coming from a quarry located in the province of Lecce (South of Italy), with total porosity P = 46–48% and porosity accessible to water $P_{H2O}$ = 39% [55] was used as the substrate to investigate the water imbibition behavior and the changes in the porous structure caused by the coatings. Indeed, Lecce stone is an adequate model of porous material with macroscopically homogeneous pore structure, but with a crucial range of pore and pore channel sizes (Figure 1) [51].

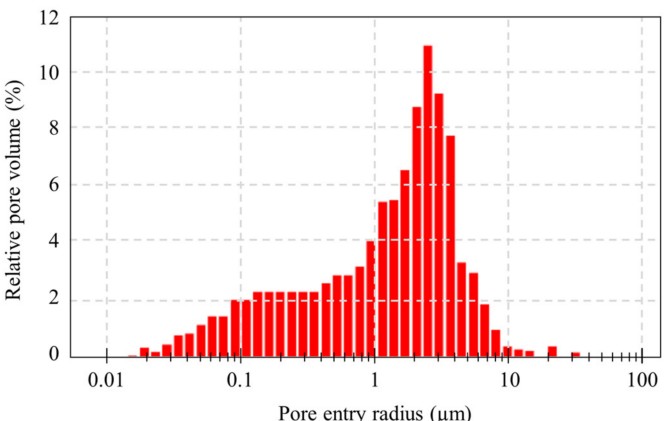

**Figure 1.** Pore size distribution of Lecce stone determined by mercury intrusion porosimetry.

The coatings were applied on only one 5 × 5 cm$^2$ surface of the prismatic stone specimens (5 × 5 × 2 cm$^3$) by deposing their solution/suspension through a pipette [36]. A mixture of 2-propanol:$H_2O$ (70:30, *w/w*) was used as a solvent for SC2-PFPE, and ethyl acetate for N215. The concentration of N215 was 1% (*w/w*). Three specimens for each treatment were prepared: one was used for MRI and NMR-MOUSE measurements, one for the photo-stability test and the third one left as reference. After solvent evaporation at room conditions, the specimens were dried in desiccator until constant mass (dry mass) was reached, and then the mass of applied coating was determined (Table 1).

**Table 1.** Mass of coating applied on Lecce stone specimens and kind of investigation subjected by the sample.

| Specimen # | Coating | Applied Coating Mass (g/m$^2$) [1] | Investigation |
|---|---|---|---|
| APL1 | SC2-PFPEsol | 16.4 | NMR |
| APL15 | | 12.4 | UV |
| APL4 | | 14.0 | reference |
| APL2 | SC2-PFPEsusp | 16.3 | NMR |
| APL18 | | 12.8 | UV |
| APL8 | | 16.0 | reference |
| APL6 | N215 | 13.6 | NMR |
| APL21 | | 11.2 | UV |
| APL14 | | 10.4 | reference |
| APL16 | NT [2] | / | NMR |
| APL12 | | / | UV |
| APL17 | | / | reference |
| APL26 | | / | reference |

[1] Accuracy of measurement: $\pm$ 0.4; [2] uncoated.

## 2.2. Coating Performance Evaluation

### 2.2.1. Photo-Stability

A xenon test chamber (Solarbox 3000 E, CO.FO.ME.GRA, Milan, Italy) equipped with a xenon lamp ($\lambda > 280$ nm) was used for inducing photo-oxidation reactions on the fluorinated compounds. The lamp worked with an irradiance of 500 W/m$^2$ and the Black Standard Thermometer was maintained at 40 °C. To evaluate the influence of the PFPE chain on the stability of the oligoamide, the precursor of SC2-PFPE (SC2, Scheme 1) was tested for comparison. Stone specimens and glass slides coated with the fluorinated compounds, and film of the fluorinated and not fluorinated products cast on KBr windows were subjected to UV irradiation. The deposition of a film on a KBr window makes possible qualitative and quantitative monitoring by FT-IR of the chemical changes during UV irradiation. The photo-stability, after 250 and 500 h of UV irradiation, was evaluated by checking the solubility of irradiated coatings, by collecting the FT-IR spectra of the films on KBr window, and by evaluating the water uptake and chromatic changes of the stone specimens.

**Scheme 1.** Synthetic route of SC2-PFPE.

The FT-IR spectra were recorded using a Perkin-Elmer spectrometer, mod. System 2000 operating in transmission mode. The working spectral range was from 370 to 4000 cm$^{-1}$, with a resolution of 2 cm$^{-1}$ and 4 scans.

The capillary water absorption test, according to UNI-EN 15801-2010 [56], was exploited to evaluate the water inhibition efficacy (*WIE*). *WIE* was calculated from the mass of liquid water absorbed by the 5 × 5 cm$^2$ surface in 30 min and 1 h, before ($A_0$) and after ($A_1$) treatment or after UV irradiation, and expressed as:

$$WIE = \frac{A_0 - A_1}{A_0} \cdot 100 \ (\%) \tag{1}$$

$A_0$ and $A_1$ are the average values calculated from the data found in three consecutive tests performed after drying the specimens before each measurement until constant mass (typically 1 week).

The chromatic features of the coated stone surface were measured by a CM-700d Spectrophotometer (Konica Minolta, Sakai, Japan) according to the procedure reported in UNI-EN 15886-2010 [57]. The data were reported in the CIE-Lab* system (CIE 1976) (Specular Component Excluded, SCE) as the average of the Lab* values obtained from three measurements of brightness (*L**) and chromatic values (*a** and *b**). The measurements performed on each specimen, before and after treatment and after UV irradiation, were carried out on the same area by using a positioning mask. The total color change ($\Delta E^*$), relative to the specimen before treatment, was calculated according to Equation (2):

$$\Delta E^* = \sqrt{\Delta L^{*2} + \Delta a^{*2} + \Delta b^{*2}} \tag{2}$$

$\Delta L^*$, $\Delta a^*$, and $\Delta b^*$ were calculated as:

$$\Delta X^* = X_u - X_c \tag{3}$$

where $X_u$ and $X_c$ are the values for uncoated (before treatment) and coated/UV irradiated specimens, respectively.

### 2.2.2. Sorptivity (MRIsorp)

Water imbibition in coated and uncoated stone specimens was carried out by magnetic resonance imaging. MRI images were acquired using Artoscan (Esaote S.p.A., Genova, Italy), a tomograph based on a 0.2 T permanent magnet, operating at around 8 MHz for $^1$H nuclei. To keep the image acquisition time shorter, a saturated acquisition approach was used, in which the $T_E$ repetition time is much shorter than the $T_1$ relaxation time. Single-slice saturated spin-echo sequences lasting 30 s were used to follow the rise of the capillary front, setting Echo time $T_E$ = 10 ms, Repetition time $T_R$ = 120 ms, a field of view of 13 × 13 cm$^2$, a slice thickness of 1 cm, determining a voxel size of 0.5 × 0.5 × 10.0 mm$^3$, large enough to reach a good signal-to-noise ratio in the reconstructed image. The stone specimens were put in absorption, inside the tomograph coil, following the same procedure of the capillary water absorption test. MRI images were acquired at increasing intervals of time, from a few seconds up to a few hours, during capillary water absorption, and then analyzed by in-house software to accurately identify the height reached by the wetting front at each measurement time. The procedure is well described in [50]. By plotting the front height (in mm) vs. the square root of water absorption time (*t*, in seconds), the absorption kinetics was fitted with both the Washburn (Equation (4)) and the modified Washburn models (Equation (5)).

In particular, Equation (5) was used to fit the front advancing when the absorption was performed from the treated face.

$$z(t) = S \sqrt{t} \quad (mm) \tag{4}$$

$$z(t) = S' \left( -\sqrt{t'} + \sqrt{t' + t} \right) \quad (mm) \tag{5}$$

where $t'$ is a new parameter, with the physical dimensions of a time, and S' can be considered as an effective sorptivity. Moreover, in the discussion of the sorptivity measurements, a $t_{0eff}$ parameter was also introduced. $t_{0eff}$ is the effective delay time deduced from the initial portion of the curves when the wetting front has not reached 0.5 mm.

### 2.2.3. Indirect Visualization of the Coating Distribution (MRIvisual)

The same tomograph Artoscan used for the sorptivity measurements was exploited to monitor the spread of water absorbed and indirectly locate the distribution of the hydrophobic treatments inside the specimens. As in the case of sorptivity measurements, the water capillary absorption experiment was performed both from the not treated and treated faces following the procedure reported in [58]. Multi-slice interleaved Spin Echo sequences of 72 s total duration were used, setting $T_E$ = 10 ms, $T_R$ = 900 ms, voxel size = $0.78 \times 0.78 \times 5.0$ mm$^3$, FOV = $20 \times 20$ cm$^3$ and collecting 13 slices per specimen (n° of scans = 8). The MRI acquisitions were performed during the water absorption process at 1, 2, 4, 24, 48, 72, 96 and 192 h from the beginning of absorption. Before each MRI acquisition, the specimens were weighted to monitor the absorbed water mass.

### 2.2.4. Porous Structure Modification

An NMR single-sided device (MOUSE PM-25; $B_0$ = 0.3 T; $G$ = 6.6 T/m) was used to acquire the transverse relaxation decay of $^1$H nuclei of water in fully saturated stone specimens. Signal intensity profiles were obtained by acquiring and averaging 20 CPMG echo intensities (3°–22°) along the shortest side of the specimens. CPMG parameters were set as following: pulse length = 8.5 μs, $T_E$ = 60 μs, $T_R$ = 2 s, n° of scans = 128, step size = 100 μm and depth range = 24 mm.

2D-NMR experiments were also performed by using stimulated spin echo (SSE) + CPMG sequences to encode diffusion–relaxation ($D$-$T_2$) parameters. In confined systems, the diffusion path of the water molecules is restricted by the pore walls, and the observed diffusion coefficient $D_{eff}$ is reduced compared to the self-diffusion coefficient $D_0$ of bulk water. $D_{eff}$ depends on the ratio between the mean squared displacement $l$ and the length scale of the confining system, i.e., the pore size, and consequently depends on the observation time of the diffusion process $\Delta$. As shown in [59] the attenuation of the diffusion coefficient $(D_{eff}(\Delta)/D_0)$ can be related to the geometry of the porous structure $(S/V)$. For short observation times and assuming pores with spherical shape [60], the average value of $S/V$ was estimated from the following equation:

$$\frac{D_{eff}(\Delta)}{D_0} = 1 - \frac{4}{9\sqrt{\pi}} \frac{S}{V} \sqrt{D_0 \Delta} \qquad (6)$$

The 2D NMR measurements were performed on fully saturated specimens at a depth of 15 mm from the treated face for $\Delta$ = 4, 12, 50, 100, 150 and 200 ms. (SSE: $\delta_{\min}$ = 0.03 ms, $\delta_{\max}$ = 0.5 ms, n° of $\delta$ = 24; CPMG: $T_E$ = 60 μs, n° of echoes = 1024, n° of scans = 64, $T_R$ = 4.5 s). The acquired NMR signal was inverted into $D$-$T_2$ correlation using the tool described in [61].

### 2.2.5. Wet-Dry Aging

The wet-dry aging consists in combining, in different ways, the conditions reported in Table 2. After each test, the wet specimens were dried at room conditions and desiccators until a constant weight was reached. After the last wet-dry cycle, all the specimens were conditioned in the dark under laboratory conditions (about 25 °C and 60% RH) for about 2 years, reaching a period of about 3 years from the beginning of the experiment. The *WIE* after 3 years, $WIE_{3\,years}$, was evaluated on each stone specimen differently aged and compared with that before aging, or time zero, $WIE_{t=0}$. The effect of aging ($AE$) was calculated using Equation (7):

$$AE = WIE_{t=0} - WIE_{3\,years} \qquad (7)$$



**Table 2.** Conditions adopted for the aging through wet-dry cycles.

| Aging Methodology | Abbreviation |
|---|---|
| 3 initial water capillary absorption tests up to 1 h | Abs |
| 6 water capillary absorption tests after UV irradiation up to 1 h | UV |
| 2 water capillary absorption tests up to 7 days for MRI then 2 saturation under vacuum tests and 4 water capillary absorption tests up to 1 h | MR |

## 3. Results and Discussion

### 3.1. Photo-Stability

The stability of SC2-PFPE to a sun-like irradiation source was evaluated by both FT-IR spectroscopy, monitoring the chemical structure of the molecules, and solubility tests. The photo-stability of the not fluorinated oligo(ethylenesuccinamide) (SC2) was also investigated by FT-IR to assess the influence of the fluorinated chain. The photo-stability of SC2-PFPE was also evaluated by determining possible changes of the protective efficacy by capillary water absorption tests, and by the color test, to determine the chromatic parameters probably associated with changes of both chemical and physical properties.

The FT-IR spectra, collected before and after UV irradiation on the film of SC2-PFPE and SC2, are reported in Figure 2. SC2-PFPE did not show evident chemical transformations on the molecule, also in the region of the amide I band (1620–1710 $cm^{-1}$) (inset of Figure 2a), the most UV sensitive functional group. On the contrary, SC2 showed modifications both in the region of the amide I and –$CH_2$- vibrations (1450–1400 $cm^{-1}$, 3000–2900 $cm^{-1}$) (Figure 2b) due to oxidation of the methylene group adjacent to nitrogen and formation of acid and aldehyde groups [62]. The photo-stability of SC2-PFPE was also supported by the solubility test. The product showed complete solubility after both 250 and 500 h of UV irradiation. For the solubility test, the same solvent employed for the deposition of SC2-PFPE on the glass slides (2-propanol) was used.

The effect of UV irradiation on the water uptake of stones, both treated and untreated, was evaluated by the analysis of *WIE* after different times of irradiation. The results are reported in Table 3. Lecce stone specimens treated with SC2-PFPE (both as solution and suspension) showed the best results, maintaining the *WIE* values higher than 90% even after 500 h of UV irradiation. On the contrary, the specimen treated with the fluoroelastomer N215 showed an anomalous behavior: the *WIE* decreased of ~60% after the first 250 h of UV irradiation, then it increased by ~220% after further 250 h of irradiation, reaching a value higher than before any irradiation (Table 3). This behavior can be explained by the rearrangement of the polymer distribution on the surface, probably due to the low adhesion to the stone (weak dipolar interaction between the coating and the rock), rather than to the degradation of the polymer. No evidence of polymer degradation was observed, and the FT-IR spectra and the solubility of the fluoroelastomer were unchanged after irradiation. Moreover, the low glass transition temperature (Tg < −10 °C) of N215 could have favored its mobility during irradiation due to temperature increase. On the other hand, N215 shows lower reproducibility of the measurements (standard deviation up to 20%) compared to SC2-PFPE (sd ≈ 1%), and a larger decrease of *WIE* from 30 min to 1 h. The low reproducibility of the *WIE* measurements for N215 may be due to an inhomogeneous distribution of the fluoroelastomer on the stone surface and/or inside the pores, which can favor the partial detachment of the polymeric film during the wetting phase, followed by its rearrangement during the drying step. The uncoated specimen also shows low reproducibility in the *WIE* data, but the effect is mainly observed for measurements after 250 h of UV irradiation. This may be explained by a modification of stone porosity due to the wet-dry aging, which can be responsible for dissolution, migration and re-crystallization of more or less soluble components of the stone. This porosity modification can change the time necessary for reaching a water absorption equilibrium (longer than the considered 30 min −1 h).

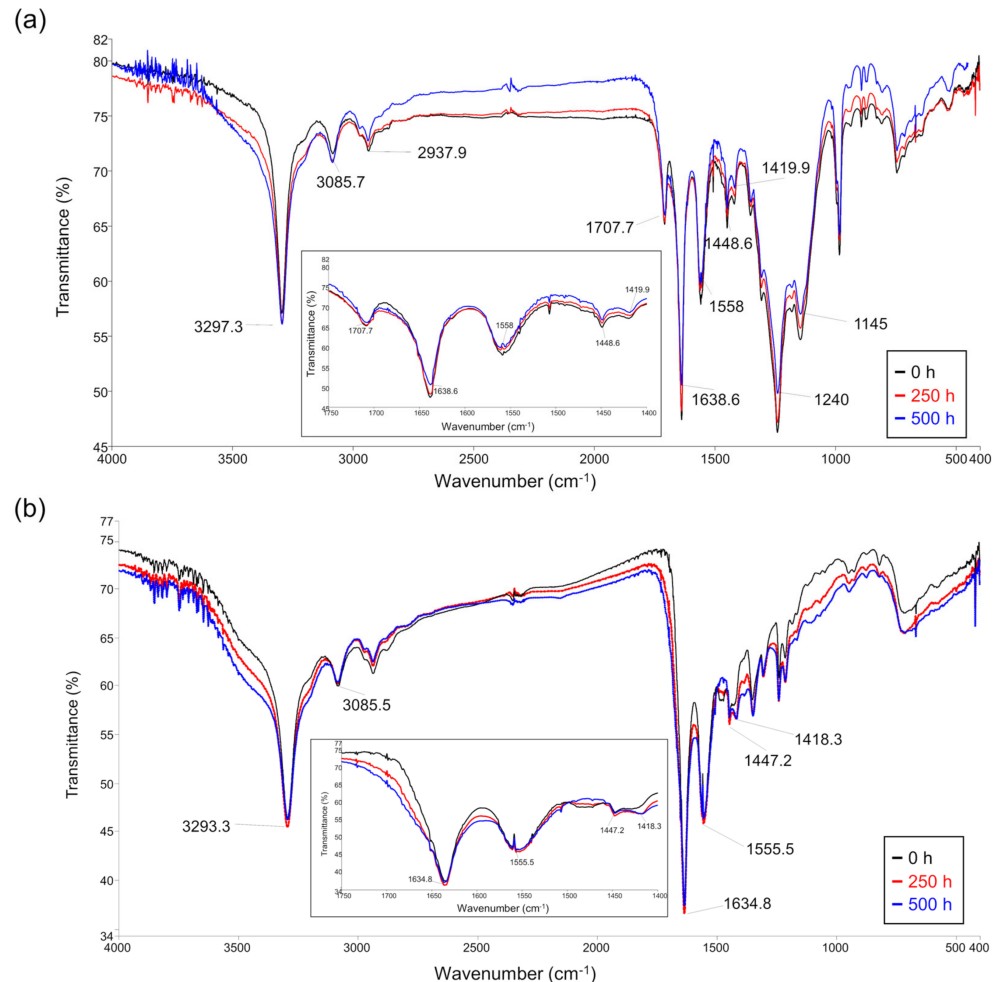

**Figure 2.** FT-IR spectra before and after different times of UV irradiation of (**a**) SC2-PFPE and (**b**) SC2. In the insets, a detail of the 1750–1400 cm$^{-1}$ region is reported.

**Table 3.** *WIE* of fluorinated coatings on Lecce stone specimens after different times (t) of UV irradiation. *WIE* was determined after 30 min and 1 h of capillary water absorption. NT means uncoated specimen.

| Specimen # | Treatment | *WIE* (%) at | | | | | |
| | | t = 0 | | t = 250 h | | t = 500 h | |
| | | 30 min | 1 h | 30 min | 1 h | 30 min | 1 h |
| APL15 | SC2-PFPEsol | 95.5 ± 0.4 | 94.5 ± 0.6 | 92.6 ± 1.2 | 90.9 ± 1.2 | 92.6 ± 0.4 | 91.7 ± 0.6 |
| APL18 | SC2-PFPE susp | 94.6 ± 1.0 | 94.7 ± 0.9 | 93.8 ± 0.5 | 93.4 ± 0.9 | 92.1 ± 0.3 | 90.9 ± 1.0 |
| APL21 | N215 | 35.2 ± 19.5 | 29.9 ± 17.0 | 13.8 ± 5.0 | 10.5 ± 4.5 | 45.1 ± 14.4 | 47.5 ± 20.1 |
| APL12 | NT | 8.6 ± 3.2 | 6.5 ± 2.2 | −7.9 ± 10.5 | −11.4 ± 9.3 | −32.1 ± 5.6 | −26.7 ± 1.1 |

Despite the inorganic composition of the stone, which components are not subjected to photo-aging, the water uptake of the untreated specimen (NT) increased with the increasing of UV irradiation times (Table 3). This behavior can be explained with the same processes of dissolution, migration and re-crystallization of the soluble components of the stone. In previous work we demonstrated that repeated wet-dry cycles, such as those occurred during the capillary water absorption test, can act as degradation factors [36], and the effect was more evident in untreated specimens than in the treated ones. To consolidate this

hypothesis, the *WIE* values reported in [36], both at 30 min and 1 h, are implemented with the data obtained in this study (Table 4).

**Table 4.** *WIE* of fluorinated coatings on Lecce stone specimens after different wet-dry aging conditions. *WIE* was determined after 30 min and 1 h of capillary water absorption.

| Specimen # | Treatment | Wet-Dry Cycles | *WIE* (%) at | | | |
|---|---|---|---|---|---|---|
| | | | t = 0 | | t = 3 Years | |
| | | | 30 min | 1 h | 30 min | 1 h |
| APL1 | | Abs + MR | 95.2 ± 0.7 | 93.3 ± 0.9 | 64.0 ± 0.9 | 57.0 ± 0.9 |
| APL15 | SC2-PFPEsol | Abs + UV | 95.5 ± 0.4 | 94.5 ± 0.6 | 68.8 ± 1.8 | 63.6 ± 0.3 |
| APL4 | | Abs | 90.9 ± 2.2 | 88.3 ± 1.9 | 89.3 ± 2.2 | 85.1 ± 3.3 |
| APL2 | | Abs + MR | 96.9 ± 0.1 | 96.5 ± 0.2 | 51.7 ± 1.7 | 45.6 ± 0.9 |
| APL18 | SC2-PFPEsusp | Abs + UV | 94.6 ± 1.0 | 94.7 ± 0.9 | 80.9 ± 3.9 | 76.9 ± 3.0 |
| APL8 | | Abs | 97.0 ± 0.2 | 96.5 ± 0.3 | 96.3 ± 0.6 | 95.3 ± 0.7 |
| APL 6 | | Abs + MR | 70.7 ± 6.3 | 64.3 ± 4.8 | 25.2 ± 9.9 | 20.0 ± 9.7 |
| APL21 | N215 | Abs + UV | 35.2 ± 19.5 | 29.9 ± 17.0 | 48.9 ± 6.4 | 48.3 ± 5.0 |
| APL14 | | Abs | 53.5 ± 4.6 | 48.1 ± 4.2 | 40.1 ± 1.5 | 35.9 ± 2.6 |
| APL16 APL12 | | Abs + MR | 0.0 ± 4.7 | −0.8 ± 2.7 | −82.1 ± 12.8 | −46.8 ± 1.3 |
| APL17 | | Abs + UV | 8.6 ± 3.2 | 6.5 ± 2.2 | −40.4 ± 8.8 | −28.8 ± 2.3 |
| APL26 | NT | Abs | 7.2 ± 3.2 | −2.5 ± 1.6 | −31.8 ± 1.9 | −6.4 ± 0.9 |
| | | Abs | 13.5 ± 4.5 | 10.1 ± 4.0 | −27.5 ± 3.5 | −11.2 ± 1.2 |

*WIE* at t = 0 corresponds to *WIE* at time zero (after treatment and before aging); *WIE* at t = 3 years is *WIE* after 3 years from the beginning of the experiment. The water capillary absorption tests carried out before the treatment are not considered because all the specimens underwent the same number of absorption tests.

The effect of aging (*AE*) is diagrammed in Figure 3. For this evaluation, we distinguished the effects on specimens subjected to different previous aging methodologies according to Table 2, opportunely combined. Abs+MR and Abs+UV in Figure 3 and Table 4 refer to specimens subjected to protracted contact with water in the case of MR conditions, and capillary tests after UV aging, respectively. Abs means only 3 tests of water capillary absorption. A drastic increase of water uptake, both after 30 min and 1 h, with negative values in the hypothetical *WIE* of the uncoated specimens, was observed for Abs+MR or Abs+UV (Table 4). For a smaller number of wet-dry cycles (Abs), an increase of water uptake is also observed, but it is less evident than in the previous cases and it is mainly limited to short times (30 min). Indeed, the *AE* at 30 min ranges from about 40, in the case of few wet-dry cycles (Abs), to 82 for Abs+MR, while the *AE* at 1 h is lower than 20 for Abs and 46 for Abs+MR (Figure 3).

On the other hand, the treated specimens, in the same conditions, generally show a lower aging effect than the untreated specimens. The *AE* for the treated specimens subjected only to Abs is very low (from 1 to 3 for SC2-PFPE and 12–13 for N215), both at 30 min and 1 h. The higher *AE* for N215 in respect to SC2-PFPE is justified by its lower hydrophobicity, and therefore to its higher water uptake during the absorption tests (Abs) (Figure S1a). On the contrary, in the specimens subjected to more drastic wet-dry aging conditions (Abs+MR), the *AE* is higher for all the treatments. However, the *AE* values at 30 min are much lower than in the untreated specimen (31 and 45 against 82), but after 1 h of water capillary absorption, similar *AEs* are observed for treated and untreated specimens, except for SC2-PFPEsol that shows the lower *AE* value (36 against 51 for SC2-PFPEsusp, 44 for N215 and 46 for the untreated specimen). This behavior can be explained by hypothesizing the different wettability of the stone in depth. For SC2-PFPEsol, which was supposed to have better penetration and distribution than SC2-PFPEsusp and a higher hydrophobicity than N215, the degradation process caused by protracted contact with water (Abs+MR) is less effective than in the other two treatments, where the hydrophobic product acts mainly at the surface without protecting the stone in depth.

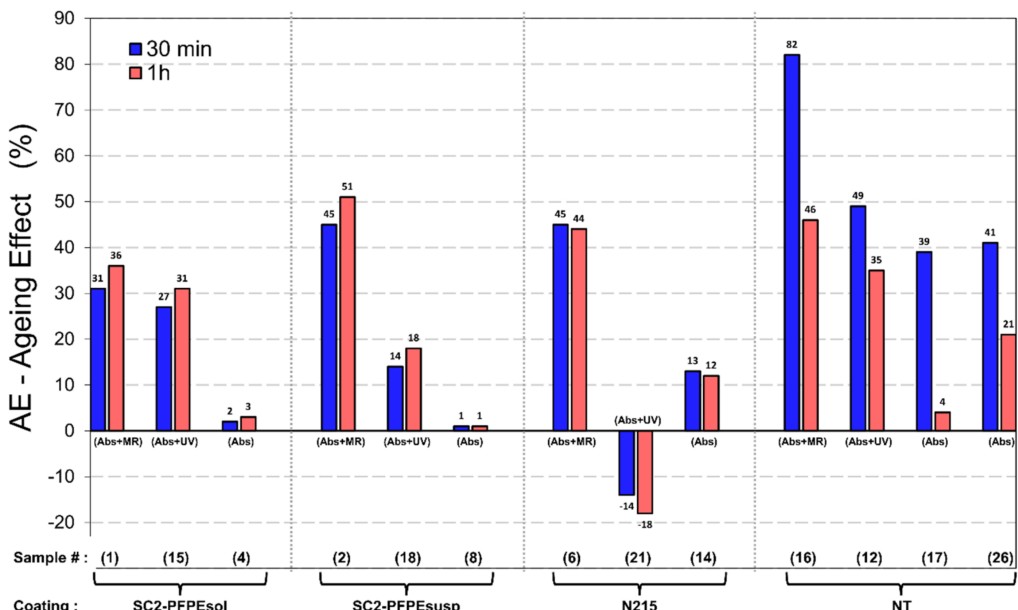

**Figure 3.** Aging effect on coated and uncoated Lecce stone specimens.

In the case of wet-drying aging associated with UV irradiation (Abs+UV), the untreated specimen again shows the higher values of *AE* (49 and 35), while N215 gives negative values (−14 and −18), indicating an improvement in the low *WIE* value found before aging. This result is in accordance with a rearrangement of the polymer film distribution on the surface, as discussed above, and here also justified by the temperature reached under irradiation (40 °C) which is higher than the Tg of fluoroelastomer polymers (<−10 °C). The same effect of rearrangement may have occurred for SC2-PFPE, a white waxy product. In this case, however, an additional factor may contribute to the aging effect. This factor is related to the chemical structure of the oligomer, which is formed by a polar chain, –CO–NH– groups able to interact by hydrogen bonds with the stone [28], and a hydrophobic segment with low interaction with stone. When water enters in contact with the treated stone for long times, the interaction coating-stone may be compromised [21], and intramolecular hydrogen bonds may be formed in the oligomer. The product, no more "well-fixed" on the substrate and in consequence of its low molecular weight (Mw 2050 instead of 125,000 for N215) can slowly diffuse into the porous specimen, depleting the surface. This may be the reason why the specimens treated with SC2-PFPE maintained their high *WIE* after 500 h of UV irradiation (Table 4), but after 3 years (with the same number of wet-dry cycles) the *WIE* was reduced. Following this hypothesis, the effect due to the penetration is expected and found more evident for SC2-PFPEsol, where a lower mass of the product is concentrated on the surface, than for SC2-PFPEsusp. Indeed, the *AE* for SC2-PFPEsol is 27 and 31 compared with the values for SC2-PFPEsusp that are 14 and 18, respectively at 30 min and 1 h.

The chromatic changes of stones induced by the coating, both after treatment and after UV irradiation, are shown in Figure 4. The chromatic variation of the stone surface after the application of the coating was imperceptible for all the products. Indeed, $\Delta E^*$ was always lower than the naked eye detection limit, ($\Delta E^* = 3$) (Figure 4a, Figure S1b).

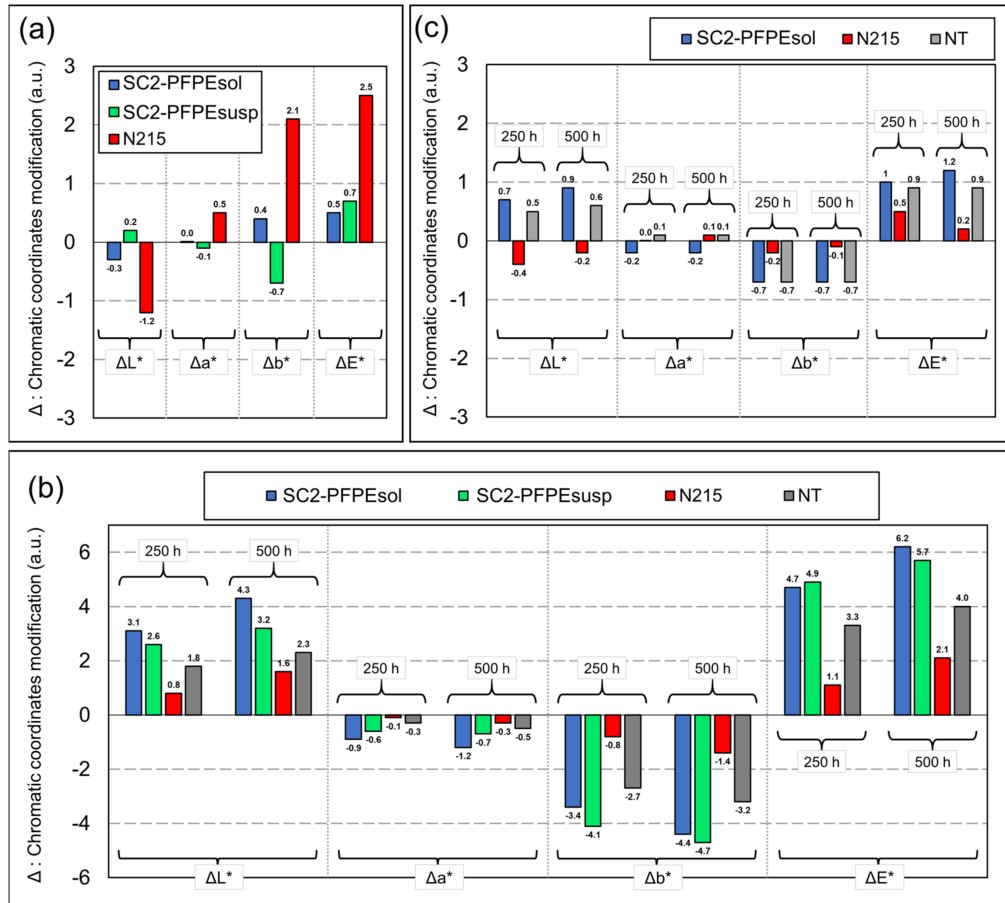

**Figure 4.** Chromatic modifications of coated stone surfaces: (**a**) after coating on Lecce stone, (**b**) after UV aging on Lecce stone, (**c**) after UV aging on marble. The variation of all the chromatic parameters refers to the same surface before coating.

Unexpected behavior of the chromatic parameters for coated Lecce stone was observed after UV irradiation (Figure 4b). Contrary to N215, for SC2-PFPE coated specimens a decrease in $L^*$ ($\Delta L^*$ increase) and an increase in b* ($\Delta b^*$ decrease) was observed, which caused an increase in $\Delta E^*$. After 500 h of UV irradiation, $\Delta E^*$ reached the values of 6.2 and 5.7 for SC2-PFPEsol and SC2-PFPEsusp, respectively. However, it must be noticed that the uncoated Lecce stone suffered the same color variation under irradiation, and a $\Delta E^* = 4.0$ was found after 500 h (Figure 4b). A color decay of stone, caused by water and UV, rather than the degradation of the coating was supposed. This hypothesis was supported by the chromatic investigations on marble specimens. Indeed, $\Delta E^*$ values around 1 were found after the same UV irradiation conditions (Figure 4c).

### 3.2. Sorptivity

The tendency of natural and artificial rocks (as porous materials) to absorb water by capillarity can be well described by the sorptivity parameter defined by Equation (4), and by Equation (5) for the effective sorptivity. Indeed, the measurement of sorptivity quantifies the velocity of water-front advancement rather than the total amount of water absorbed and it also provides a measure of how the pore size affects the capacity of absorption. The accuracy of the measurement depends on the detection method used to collect data. Contrary to the gravimetric method, the monitoring of the dynamic process of water transport in the porous medium by MRI allows the observation of the displacement of water and to determine the penetration depth. An example of the evolution of water uptake monitored by the MRIsorp procedure is shown in Figure 5. The presence of water inside the stone specimen is evidenced by the bright area.

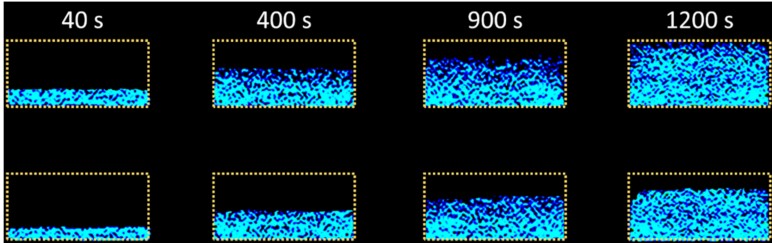

**Figure 5.** Evolution over time of the water uptake in the uncoated Lecce stone specimen (NT) (APL 16) (top), and the N215 coated specimen (APL 6) (bottom) monitored by MRIsorp procedure. The capillary water absorption for the coated specimen was carried out through the uncoated face.

The height reached by the wetting front at each acquisition time was identified by in-house software, according to [50]. The relationship between the height of the wetting front and the absorption time (*t*) for uncoated and coated specimens is shown in Figures 6 and 7. Figure 6 relates to the absorption from the untreated face, while Figure 7 relates to the absorption from the treated one. When the water absorption was performed through the untreated face of the coated specimens, the rising of the wetting front vs. the square root of absorption time followed a linear trend, as for the uncoated specimen (NT) (Figure 6). The Washburn model (Equation (4)) was used in all cases for computing the sorptivity (S). On the contrary, the rising of the wetting front through the treated face was characterized by a different behavior, that could be well modeled by introducing the parameter (*t′*), by the use of Equation (5), allowing the computation of the effective sorptivity S′ (Figure 7). To compare the sorptivity from the absorption through the treated face with that from the untreated face, the *S* parameter for the absorption from the treated face was computed by considering only the points acquired at the longest times, where the trend of the curve was linear (green linear fitting in Figure 7).

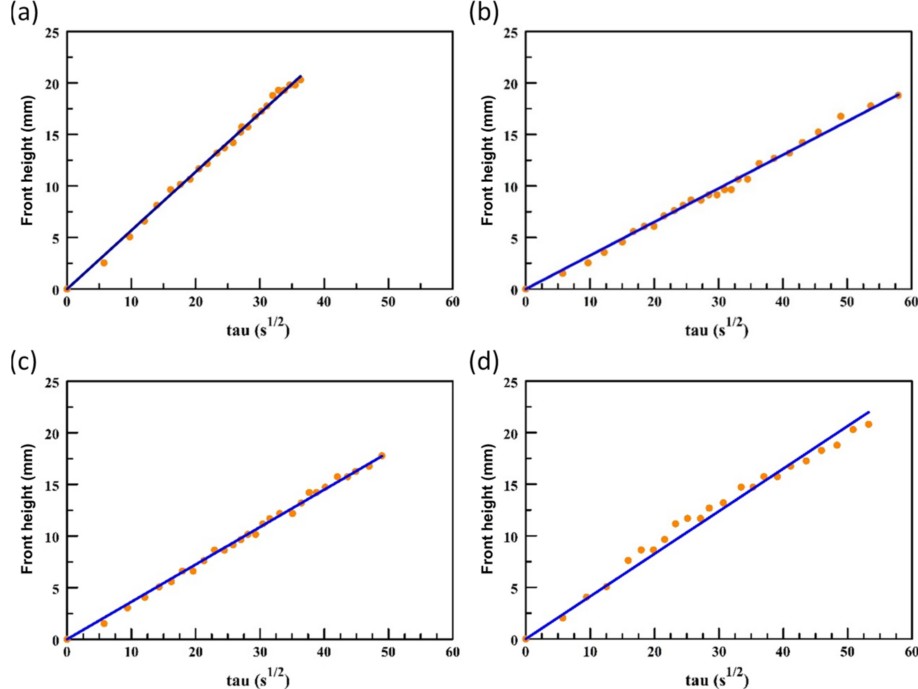

**Figure 6.** Height of the wetting front in uncoated and coated Lecce stone specimens for absorption through the untreated face vs. the square root of time of water absorption. (**a**) Uncoated (APL16); (**b**) FSC2-PFPEsol coating (APL1); (**c**) FSC2-PFPEsusp coating (APL2); (**d**) N215 coating (APL6). The blue straight line represents the fit by Equation (4); tau is the square root of the acquisition time.

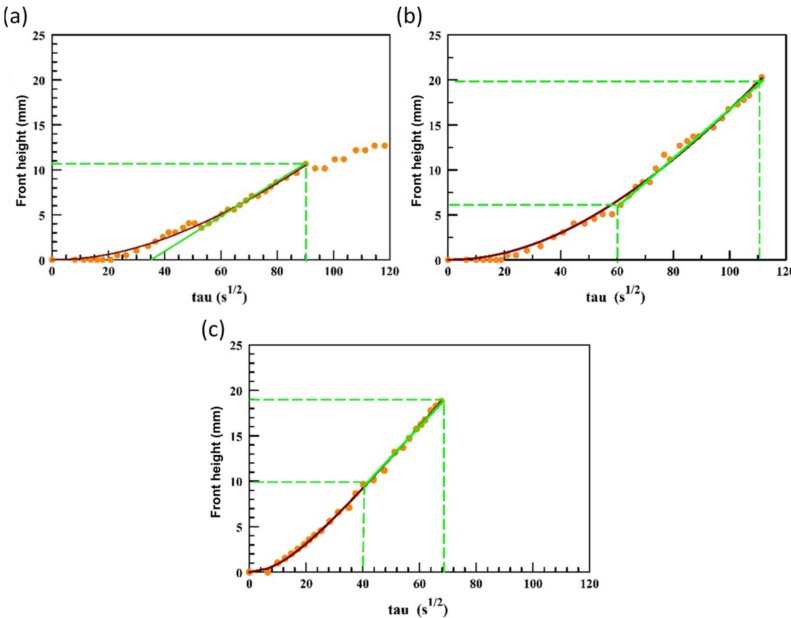

**Figure 7.** Height of the wetting front in coated Lecce stone specimens vs. the square root of time of water absorption through the treated face. (**a**) FSC2-PFPEsol coating (APL1); (**b**) FSC2 PFPEsusp coating (APL2); (**c**) N215 coating (APL6). Green lines refer to the linear fits by Equation (4) of the data points on the longest times of the curve; Red lines refer to the fits by Equation (5).

Due to the presence of the coating, both a reduction in pore size and a change in stone wettability is expected, with an unsurprising decrease in the sorptivity associated with an increase in the effective delay time of the front height ($t_{0eff}$). Indeed, the $S$ values found for the absorption through both the untreated and treated face were lower than that of the uncoated specimen (Table 5), and SC2-PFPE showed a higher decrease than N215. At the same time, SC2-PFPEsol provided lower sorptivity than SC2-PFPEsusp and, consequently, higher $t_{0eff}$. Additionally, contrary to the other specimens, the wetting front of SC2-PFPEsol rose quickly until 10 mm and then the velocity decreased showing an asymptotic behavior up to around 13 mm. For this specimen (Figure 7a) the data were fit up to tau = 90. A similar trend was found for the values of the effective sorptivity (S') and the $t'$ parameter, except for the surface coated with SC2-PFPEsusp (Table 5). In this case, the S' and $t'$ values are higher than those of N215 and SC2-PFPEsol. For the sake of clarity, S' and $t'$, unlike $S$, are mainly determined by the transport properties of the coating layer. Therefore, the higher values of S' and $t'$ for SC2-PFPEsusp, compared with N215 and SC2-PFPEsol, can be explained with an accumulation of the highly hydrophobic coating on the surface of the treated face, which caused an initial dramatic water diffusivity reduction followed, at a long time, by a fast rise of the front height.

**Table 5.** Sorptivity for coated and uncoated Lecce stone specimens.

| Specimen | Coating | Untreated Face | | Treated Face | | |
|---|---|---|---|---|---|---|
| | | $S$ (mm/s$^{1/2}$) | $S$ (mm/s$^{1/2}$) * | $t_{0eff}$ (s) | S' (mm/s$^{1/2}$) | $t'$ (s) |
| APL 1 | SC2-PFPEsol | 0.33 | 0.20 | 537 | 0.29 | 9166 |
| APL 2 | SC2-PFPEsusp | 0.36 | 0.28 | 440 | 0.46 | 14,533 |
| APL 6 | N215 | 0.41 | 0.32 | 99 | 0.37 | 371 |
| APL 16 | NT | 0.57 | – | | | |

* Sorptivity was computed on the linear portion of the curve on the longest times of the acquired data in Figure 7; $t_{0eff}$ = effective delay time deduced from the initial portion of the curves of Figure 7 when the wetting front was at 0 mm; $t'$ was computed from Equation (5).

To explain the results of sorptivity measurements, further considerations must be done. Firstly, different compounds exhibit different hydrophobicity, which causes an obvious difference in the wettability of the stone. Secondly, the mass of coating applied in the porous specimen is not always the same for all the treatments reducing the pore space in different ways. Finally, the molecular size of the compounds and the solubility may affect the penetration and distribution of the coating inside the stone with possible accumulation of the product in specific areas. This accumulation can create a physical barrier to water entry.

The different hydrophobicity of the coatings estimated through gravimetric data (Figure 8a) showed a modest reduction in the mass of water absorbed from the untreated face for all the coatings, even if SC2-PFPEsol reduced the absorption slightly more than N215 and SC2-PFPEsusp. On the contrary, the gravimetric data collected for the absorption through the treated face showed a drastic reduction of water uptake for the specimens coated with SC2-PFPE, but not for that coated with N215.

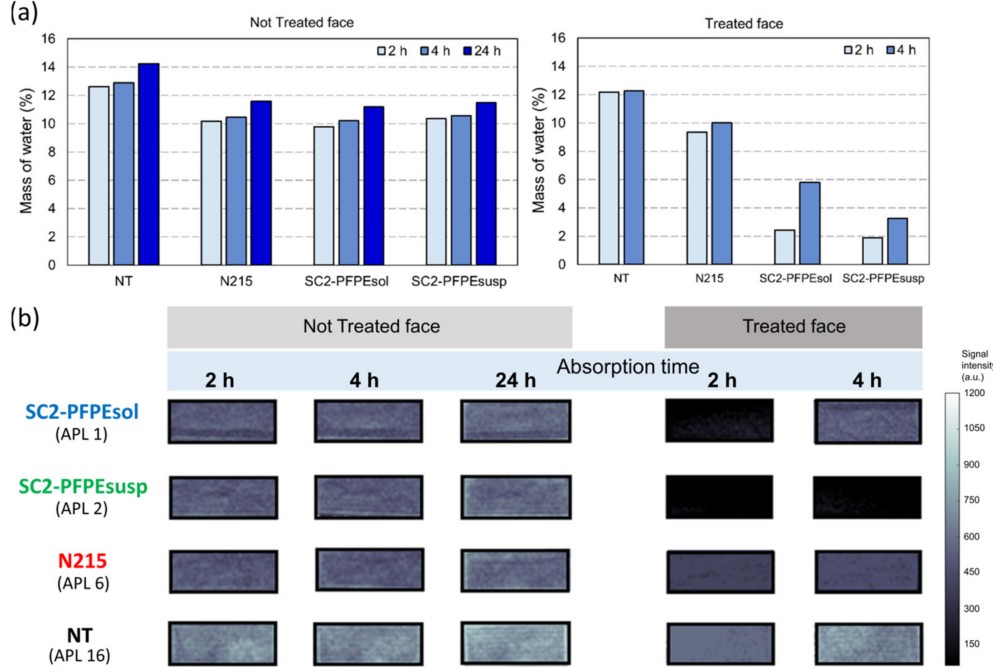

**Figure 8.** Kinetics of water absorption in uncoated (NT) and coated Lecce stone specimens. (**a**) Mass of water absorbed over time. The mass was computed as water mass absorbed per 100 g of stone; (**b**) MRI images of an internal section of the stone specimens over the water absorption time. Absorption was performed through both the untreated and treated face.

The hydrophobicity of the coatings can also be estimated by MRIvisual. This procedure was exploited to get internal sections of the stone specimens. Indeed, MRI images of internal sections of Lecce stone specimens collected at increasing times of water absorption allowed us to indirectly visualize the coating distribution in the specimens. Figure 8b shows, for the examined specimens, MRI images of an internal section of each coated and uncoated specimen, for absorption through both the untreated and treated face. The presence of the hydrophobic coating is identified by darker or less bright areas.

Based on the images of Figure 8b, concerning the absorption through the untreated face, SC2-PFPEsol gave the best penetration with an apparent accumulation of the coating at the bottom of the specimen, evidenced by the black region close to the bottom of the image. On the contrary, SC2-PFPEsusp showed a uniform water uptake indicating a preferential deposition on the treated surface, due to its partial solubility. N215, with an average molecular weight higher than SC2-PFPE (125,000 against 2050 g/mol), showed a polymer accumulation at about 5 mm from the treated face, although at 4 h of water contact

through the not treated face a darker area, with a U-shape, is observed up to a depth of about 18 mm from the treated face (Figure 8b).

As the mass of product applied is similar for all the coatings (Table 1) and not enough to drastically reduce the pore volume of Lecce stone (SI in [36]), and the high residual vapor diffusivity (>85%, [36]) proves the coating did not block the pores, SC2-PFPE was demonstrated to provide the best hydrophobic surface.

The sorptivity data are consistent with the MRIvisual and gravimetric measurements, but provide additional information otherwise not revealed. The modest reduction of $S$ and the short effective delay time ($t_{0eff}$) (Table 5) showed by the specimen coated with N215 can be mainly attributed to the modest hydrophobicity of this coating. Therefore, water is quickly absorbed and diffused as in a stone with medium and large pores, such as the untreated Lecce stone [63]. The phenomenon is not only observed for absorption through the untreated face, but also through the treated one, where a higher accumulation of N215 on the first mm was detected. This proved the absence of blocked pores, but confirmed the modest hydrophobic effect achieved with N215. The lower S values and very long $t_{0eff}$ (>500 s) for SC2-PFPEsol compared to SC2-PFPEsusp are due to the different penetration and distribution of the coating. Though the gravimetric data and the MRI images (Figure 8) showed the SC2-PFPEsusp coated face more resistant to water absorption than SC2-PFPEsol, the sorptivity through the treated face of the specimen coated with SC2-PFPEsusp (0.28 mm/s$^{1/2}$) was greater than that with SC2-PFPEsol (0.20 mm/s$^{1/2}$). This result is an evidence that the SC2-PFPEsol coating provides a slow water absorption and distribution, similarly to a porous substrate with small pores which provide a lower water absorption rate than the largest ones.

The signal intensity profile obtained on saturated stone specimens by the NMR single-sided device (Figure 9) agrees with a distribution of SC2-PFPEsol in medium-large pores, rather than in the small ones, confirming its low sorptivity. The evident decrease in signal intensity at a depth between 15,000 and 19,000 μm for the SC2-PFPEsol coated specimen agrees with the hypothesis of an accumulation of the hydrophobic product in this area and is in good accordance with the MRI images. For sake of clarity, it should be remembered that no pore blockage occurred due to the high residual vapor diffusivity, so only the coating of the pore walls is justifiable. Moreover, the coated pores can only be the medium-large ones, otherwise the signal intensity profile would be similar to that shown by SC2-PFPEsusp and N215. In the case of SC2-PFPEsusp, the not dissolved component present in the suspension makes this coating little prone to diffuse inside the stone and, although its sorptivity is lower than that of N215 (0.28 against 0.32 mm/s$^{1/2}$ for the treated face), it does not guarantee long-lasting protection against liquid water, as confirmed by the aging effect reported in Figure 3.

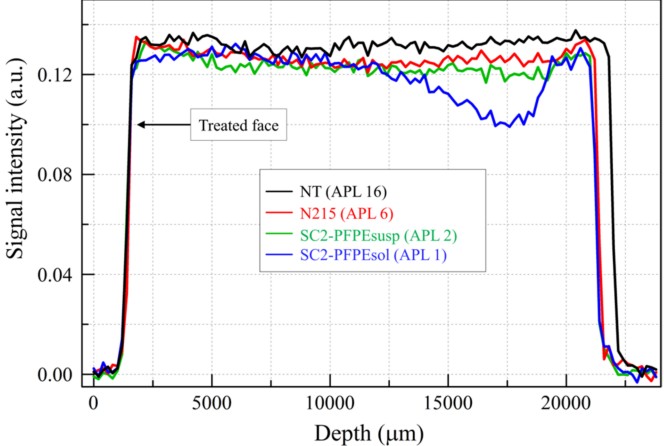

**Figure 9.** NMR signal intensity profile of water saturated coated and uncoated Lecce stone specimens acquired by NMR single-sided device.

### 3.3. Porous Structure Modification

To better characterize the porous structure modifications in the region of apparent accumulation of SC2-PFPEsol (about 15 mm from the treated face), the effective water diffusion coefficient was monitored at various observation time ($\Delta$ = 4, 12, 50, 100, 150 and 200 ms) and a comparison was made among the results obtained from the uncoated specimen and the coated ones. Indeed, the effective diffusion coefficient is related to the geometry of porous media, but not to the surface wettability as it is unaffected by the surface relaxivity. Moreover, for short observation times of the diffusion process, the observed diffusion coefficient decreases with the square root of $\Delta$ and depends on the surface to volume ratio S/V (i.e., pore size) (Equation (6)). Figure 10 shows $D$-$T_2$ correlation maps at increasing values of $\Delta$: in particular, $D_{eff}$ decreases as the observation time increases due to the confinement in the porous structure, and the average value of $T_2$ slightly increases at longer observation times. This behavior is typical for a porous medium with a wide range of pore sizes and good pore connectivity [59,60,64]. For the shortest observation times the detected NMR signal belongs to water in the whole range of pore sizes, and the value of diffusion coefficient is close to that of the bulk water (e.g., for $\Delta$ = 4 ms, $D_{eff}$ ~ 2 $\mu m^2$/ms). For the longest values of $\Delta$, $D_{eff}$ decreases and the acquired NMR signal shows an apparent shift to longer $T_2$s (water in large pores) because of the signal lowering in the smaller pores, due to their shortest $T_2$. The map from the specimen coated with SC2-PFPEsusp is similar to that from the uncoated specimen (NT). This confirms that the product was mainly concentrated at the treated surface, while the porous structure at 15 mm from the treated face was almost unchanged. This behavior is further confirmed by the estimation of S/V from the diffusion-relaxation maps through Equation (6) (Figure 11 and Table S1). For sake of clarity, the $D_{eff}$ to solve Equation (6) was estimated from the highest intensity peak of $D$-$T_2$ correlation maps and the corresponding data are reported in Table S2.

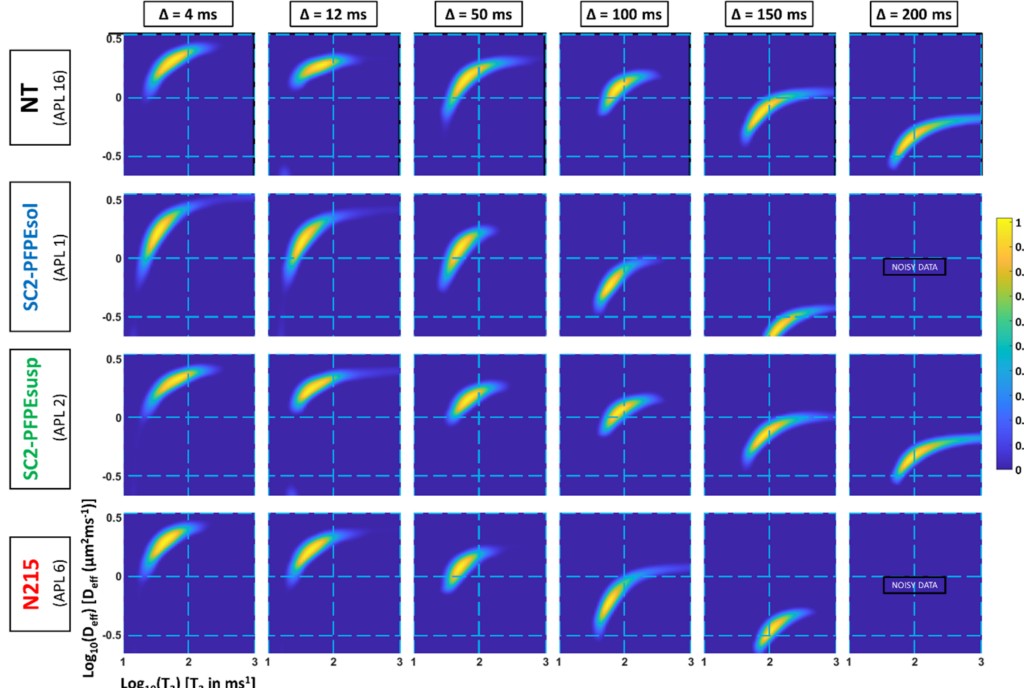

**Figure 10.** 2D maps at 15 mm depth of $Log_{10}(D_{eff})$ vs. $Log_{10}(T_2)$ for coated and uncoated Lecce stone specimens. For each computed correlation map, the signal intensity was normalized to the maximum intensity.

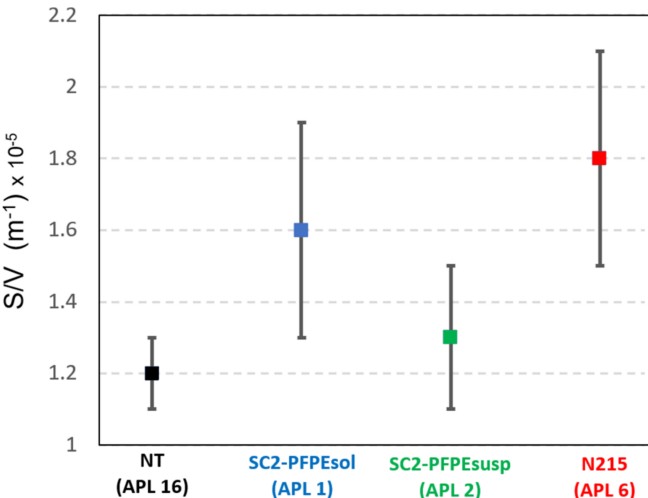

**Figure 11.** Estimation of the average S/V through the diffusion coefficient method for uncoated and coated Lecce stone specimens.

The 2D maps of the specimens coated with SC2-PFPEsol and N215 show similar behavior with a faster decrease of $D_{eff}$ compared to the uncoated and SC2-PFPEsusp coated specimens. Unlike for NT and SC2-PFPEsusp, not computable NMR signal was detected at $\Delta = 200$ ms, and for these specimens, a higher S/V was estimated (Table S1). These results suggest a decrease of the largest pores due to the presence of the coating and are consistent with the MRI images (Figure 8b). Finally, the comparable penetration depth of N215 and SC2-PFPEsol agrees with the explanation provided for the sorptivity data, where the higher sorptivity and lower delay time observed for stone coated with N215 was justified with its lower hydrophobic effect compared to SC2-PFPE.

To more easily evaluate the performance of the tested coatings, and to visualize the concordance among the results obtained with different methodological approaches, a summary of the main data is reported in Table 6.

**Table 6.** Summary of the most important properties provided by SC2-PFPE and N215 coated Lecce stone specimens. Untreated specimens (NT) were used as a reference.

| Coating | Specimen | Wet-Dry Cycles (Aging) | WIE at 30 min (%) | | | Color Change $\Delta E$ | | Sorptivity (mm/s$^{1/2}$) | S/V (m$^{-1}$) $\times 10^5$ |
| | | | After Coating | After UV Aging (500 h) | After Aging (3 Years) | After Coating | After UV Aging (500 h) | From Coated Face | At 15 mm from the Coated Face |
|---|---|---|---|---|---|---|---|---|---|
| SC2-PFPEsol | APL1 | Abs + MR | 95.2 | | 64.0 | | | 0.20 | 1.6 |
| | APL15 | Abs + UV | 95.5 | 92.6 | 68.8 | | 6.2 | | |
| | APL4 | Abs | 90.9 | | 89.3 | 0.5 | | | |
| SC2-PFPEsusp | APL2 | Abs + MR | 96.9 | | 51.7 | | | 0.28 | 1.3 |
| | APL18 | Abs + UV | 94.6 | 92.1 | 80.9 | | 5.7 | | |
| | APL8 | Abs | 97.0 | | 96.3 | 0.7 | | | |
| N215 | APL6 | Abs + MR | 70.7 | | 25.2 | | | 0.32 | 1.8 |
| | APL21 | Abs + UV | 35.2 | 45.1 | 48.9 | | 2.1 | | |
| | APL14 | Abs | 53.5 | | 40.1 | 2.5 | | | |
| NT | APL16 | Abs + MR | 0.0 | | −82.1 | | | 0.57 | 1.2 |
| | APL12 | Abs + UV | 8.6 | −32.1 | −40.4 | | 4.0 | | |
| | APL17 | Abs | 7.2 | | −31.8 | - | | | |
| | APL26 | Abs | 13.5 | | −27.5 | - | | | |

## 4. Conclusions

The new oligo(ethylensuccinamide) containing low molecular weight pendant per-fluoropolyether segments shows suitable characteristics to be used as a protective agent

for stone materials. Although its solubility in alcohols or hydro-alcoholic solvents is relatively low (0.5% *w/w*), it is enough to make possible the application (either by brush or by spray) of this coating on highly stone surfaces of interest in the field of Cultural Heritage, without any environmental negative impact. The penetration and diffusion of SC2-PFPE inside the stone, when applied as a solution, show excellent and comparable performance to N215 (commercial product), but with a higher hydrophobic effect due to the orientation of the fluorinated segments at the air/stone interface [36]. The lower sorptivity and the higher effective delay time observed for SC2-PFPEsol coated stone compared to the N215 one is a direct evidence of the advanced performance of SC2-PFPE. Moreover, the perfluoropolyether segments guarantee good stability to UV irradiation, while the not fluorinated amidic groups of the oligo(ethylensuccinamide) interact, via hydrogen bonds, with the stone giving a good permanence of the product on the surface also after aging (UV irradiation and wet-dry cycles). This implies the coating provides good reversibility, as well as durability. It is worth pointing out that the high photo-stability of the fluoroelastomer to aging, on the other hand, is not enough to assure a low wettability of stone surface when this low amount of product is applied. Indeed, although the protective efficacy of N215 can increase after aging, the values remain much lower than those of SC2-PFPE.

Last but not least, this study confirms that NMR techniques are powerful tools for monitoring the water absorption and spreading inside the stone, as well as for estimating the modification of the pore structure due to the application of the coating. The diffusometry measurements of $^1$H-nuclei of water allow the estimation of the average surface to volume ratio (S/V) (i.e., pore structure) almost independently of the surface relaxivity. These data help to clearly show the distribution of the coating in certain classes of pores otherwise not obtainable in a non-destructive way. Moreover, the good agreement between MRI results and the signal intensity profiles acquired by a portable NMR single-sided device paves the way to the use of NMR techniques directly in-situ.

**Supplementary Materials:** The following are available online at https://www.mdpi.com/article/10.3390/coatings11040452/s1, Figure S1. Photos of coated and untreated Lecce stone specimens after 3 years from the treatment and aging with 3 cycles of water capillary absorption (Abs). The specimens APL4 and APL8 were coated with SC2-PFPEsol and SC2-PFPEsusp respectively, while APL 14 with N215. APL17 is the untreated specimen: (a) The specimens after 30 min of water capillary absorption. The absorption of the specimens APL4, APL 8 and APL 14 was carried out through the coated face. The water uptake is evident in APL17 (completely wet) and APL 14 (wet up to half height), while the APL 4 and APL 8 specimens show limited water absorption; (b) the coated/untreated surface of the same specimens in dry conditions. Table S1. Estimated average surface to volume ratio for coated and uncoated Lecce stone specimens computed through Equation (6) with fit error corresponding as one standard deviation. Table S2: Effective diffusion coefficients estimated from the highest intensity peak of D-$T_2$ correlation maps and expressed as average values with uncertainties ~5% corresponding to one standard deviation.

**Author Contributions:** Conceptualization, M.C. and L.B.; methodology, M.C., V.B. and L.B.; software, V.B. and L.B.; formal analysis, A.P. and Y.C.; investigation, M.C., V.B., A.P., A.S. and L.B.; data curation, M.C., V.B. and L.B.; writing—original draft preparation, M.C.; writing—review, V.B., Y.C., A.P. and A.S.; writing—review and editing, M.C. and L.B.; supervision, M.C. All authors have read and agreed to the published version of the manuscript.

**Funding:** This research received no external funding.

**Institutional Review Board Statement:** Not applicable.

**Informed Consent Statement:** Not applicable.

**Data Availability Statement:** Not applicable.

**Acknowledgments:** Leonardo Brizi would like to thank Sabine Haber-Pohlmeier and Andreas Pohlmeier for the assistance in realizing the experiments with the NMR-MOUSE PM25 apparatus. All authors would like to acknowledge Paola Fantazzini for precious discussions on NMR data.

**Conflicts of Interest:** The authors declare no conflict of interest.

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
