# Peer review of "High Efficiency Fluorinated Oligo(ethylenesuccinamide) Coating for Stone"

_coatings, doi:10.3390/coatings11040452_

Round 1

Reviewer 1 Report

The paper is well structured and presented. The test methods are identified as well as the standards procedures. The reviewer suggests the following changes to improve the quality of the paper:

  • Based on Figure 1 results, what is the open porosity of the stone? Type?
  • In line 170 what is the criterion adopted for delta E; in line 377 values of 6.2 are referred;
  • Figures from the samples, and in the wet-dry ageing should be added; what was the capillary water absorption procedure (by immersion?); In 2.2.5 please explain better the drying periods and the position of the samples (maybe adding photo with coated and uncoated samples) for this test;
  • The dimensions of the samples for each test should be clarified; number of measurements/samples - maybe a table with the synthesis of this data would be better
  • Figure 3 should appear near the text which is mentioned
  • The natural ageing climate conditions should be better characterized during the 3 years;
  • The quality of figure 4 should be improved
  • The paper has a significant amount of results, thus the relevant results, especially when more than one type of measurement is made by different techniques, should be synthetized/discussed in a new section – “critical analysis” with table(s) for uncoated, coated, before and after ageing, highlight the main results and compared with other authors. Moreover, correlations between parameters should be discussed. If the authors could make this at the end of the paper, before the conclusions, it would improve the paper
  • Practical applications should also be highlighted on the protection of stone in Cultural Heritage; specific requirements should be discussed, for example: reversibility of the protection and its durability (photos?); the field of application should be highlighted (for example: is suitable for each type of stone; or with a range of open porosity; etc.)
  • Environmental assessment usually is carried out by Life Cycle Assessment. The authors should explain in a quantitative way which aspect is assessed to conclude “Environmental Friendly Coating”;
  • Furthermore, eco-toxicity should be mentioned, at least for further studies, highlighting the importance of testing the leachates over time during ageing.

Reviewer 2 Report

The manuscript concerns a very interesting study on testing specific organic coatings for stone treatment. The paper is very well-structured, the experimental procedure followed is intensive, while the research results are documented in detail. However, there are some general aspects that should be considered by the authors, as well some more specific comments.

The title does not accurately describe the findings of the study, since the low environmental footprint of the materials tested is not documented, whereas issues related to the compatibility of the proposed coatings with historic material is not assessed. Therefore, it is recommended to revise the title, deleting the terms ‘environmental friendly’ and ‘cultural heritage protection’.

Regarding the compatibility and effectiveness of the proposed materials in historic stones’ treatment, it is significant to present experimental data regarding the porosity and pores size alteration of the specimens near to the coating surface. An indirect assessment of this is given in some parts of the text, however there is no relevant experimental data. Additionally, the permeability of the treated specimens is a significant property to be measured, taking into account that the evaporation and circulation of air and humidity inside the structure of historic masonries are of paramount importance.

The more specific comments are as following.

Abstract

Line 30. Rephrase ‘dramatically changes’ with ‘minimizes’

Line 31. Rephrase ‘positively’ with ‘by’

Introduction

Lines 39-41. Very general assumption needing better documentation

Line 41. The characterization of stones as ‘porous’ is not accurate, since there are many types (i.e. marble) with extremely low porosity (1-2%).

Line 45. Rephrase ‘the binder’ with ‘mortars’. Binders are a constituent of mortars

Lines 51-57. Please add relevant References

Line 59. Delete ‘ideal’ since it is a rather exaggerating term

Materials and Methods

Line 16. Rephrase ‘near-ideal’ with ‘adequate’

Lines 17-18. Rephrase ‘being …homogenous’ as following ‘with macroscopically homogenous pore structure’

Line 24. Were the specimens prismatic? Rephrase the term ‘sample’ as elsewhere in the text with ‘specimen’

Line 32. Add ‘was’ before ‘determined’

2.2.5 Wet-dry ageing

The methodology followed should be better documented. Was it on a base of a relevant standard?

Results and discussion

Lines 41-45. This part should be deleted since it is a repetition

Lines 46-52. This part should be better moved to the Materials and Methods section

Lines 95-99. More documentation should be made related to pore size distribution and porosity experimental results

Page 10

Lines 53-58. Relevant references should be given

Line 58. Rephrase the term ‘no more well fixed’

Lines 70-73. Macroscopic or/and microstructural photos of the specimens should be added

Lines 82-83. Were there other stone specimens (marble) tested?

Figure 5. Is there a relevant figure with a coated specimen?

Page 12

Lines 28-30. In which depth an alteration of the pores size is expected? Relevant research data should be given

Page 13

Lines 79-82. This part is crucial and should be testified with more experimental data regarding porosity alterations

Page 14

Lines 5-10. Relevant references and better documentation are needed

Page 15

Lines 23-29. Relevant references should be added

Reviewer 3 Report

An oligo(ethylensuccinamide) is synthesized and evaluated for the protection of Lecce stone. This is a very good article, supported by experimental measurements which are not commonly used/reported in the field of stone conservation. Therefore, the NMR and particularly the MRI measurements are highly appreciated as they point towards novel scientific directions. For these reasons I believe that the manuscript should be published in Coatings. I have only some minor suggestions.

Lines 241-245: The first paragraph of the Results and Discussion should be removed or moved to the Introduction.

SC2-PFPE with concentration of 0.5% (w/w) is called solution (SC2-PFPEsol) whereas SC2-PFPE with concentration of 1% (w/w) is called suspension (SC2-PFPEsusp). Is the solubility known?

Lines 69-70: “As in the case of other polymeric materials, the big molecules can be responsible for the deposition of a superficial film with partial or total pore blockage [33].” At this point the authors should acknowledge that siloxanes do not have this disadvantage and they are alternative products for stone conservation. Moreover, some of them are environmentally friendly e.g. Macromolecular Symposia 2013, vol 331-332 (1), 158–165.

There, is an increasing concern on the effects of fluorine. If fluorine evolves from the material either as fluorine or fluoride, it may affect the natural stone and actually accelerate its degradation. A comment on this point in the Introduction will be helpful to show to the reader that the authors are aware of this tricky risk.

Reviewer 4 Report

I find this manuscript excellent and presenting innovative important results. Despite that, I have some minor remarks.

At first,  a reader might be confused with short sentences finished with a huge number of references, and his/her knowledge in this field would not be ameliorated. Therefore, I strongly suggest developing sentences based on [1-13], [12-21], [26-32] to show the real contents of cited references. No, a reader cannot be certain if this research is really novel, as well if the scientific contribution into this paper done by the authors is.

Please use proper writing of decimals: 2.13 instead of 2,130; 125 instead of 125,000 for molecular weights.

Please show how biocalcarenite has been obtained, from what source.

Please give the name of a company delivering chemicals and their purities.

Please add the reference [50] to the caption of Figure 1 (I understand that the presented picture has been taken from this source?).

Please give a space between a value and a unit: 1 h instead of 1h (line 310).

Round 2

Reviewer 1 Report

The author followed the majority of the suggestions. The paper has improved.

Reviewer 2 Report

Authors have taken into account all commnents and made respective revision in their manuscript. In its present form, the paper is very well structured with detailed research findings and could be published in the Journal.